# Dual Student Networks for Data-Free Model Stealing

**James Beetham**[1*]**, Navid Kardan**[1*]**, Ajmal Mian**[2]**, Mubarak Shah**[1]
[1]Center for Research in Computer Vision
University of Central Florida
Orlando, Florida 32816, USA
[2]Department of Computer Science
University of Western Australia
Crawley WA 6009, Australia
{james.beetham,kardan}@knights.ucf.edu
ajmal.mian@uwa.edu.au
shah@crcv.ucf.edu

## Abstract

Data-free model stealing aims to replicate a target model without direct access to either the training data or the target model. To accomplish this, existing methods use a generator to produce samples in order to train a student model to match the target model outputs. To this end, the two main challenges are estimating gradients of the target model without access to its parameters, and generating a diverse set of training samples that thoroughly explores the input space. We propose a Dual Student method where two students are symmetrically trained in order to provide the generator a criterion to generate samples that the two students disagree on. On one hand, disagreement on a sample implies at least one student has classified the sample incorrectly when compared to the target model. This incentive towards disagreement implicitly encourages the generator to explore more diverse regions of the input space. On the other hand, our method utilizes gradients of student models to indirectly estimate gradients of the target model. We show that this novel training objective for the generator network is equivalent to optimizing a lower bound on the generator's loss if we had access to the target model gradients. In other words, our method alters the standard data-free model stealing paradigm by substituting the target model with a separate student model, thereby creating a lower bound which can be directly optimized without additional target model queries or separate synthetic datasets. We show that our new optimization framework provides more accurate gradient estimation of the target model and better accuracies on benchmark classification datasets. Additionally, our approach balances improved query efficiency with training computation cost. Finally, we demonstrate that our method serves as a better proxy model for transfer-based adversarial attacks than existing data-free model stealing methods.

## 1 Introduction

Model stealing has been shown to be a serious vulnerability in current machine learning models. Machine learning models are increasingly being deployed in products where the model's output is accessible through APIs, also known as Machine Learning as a Service. Companies put a large amount of effort into training these models through the collection and annotation of large amounts of data. However, recent work has shown that the ability to query a model and get its output – *without access to the target model's weights* – enables adversaries to utilize different model stealing approaches, where the attacker can train a student model to have similar functionality to the target model (Kesarwani et al., 2018; Yu et al., 2020; Yuan et al., 2022; Truong et al., 2021; Sanyal et al., 2022). Two major motivations for stealing a private model are utilizing the stolen model for downstream adversarial attacks as well as monetary gains, therefore, model stealing methods present an increasing problem (Tramèr et al., 2016; Zhang et al., 2022).

---

*Equal contribution

Data-free model stealing is a generalization of black-box model stealing which is an extension of knowledge distillation. In all three areas, the aim is to obtain a student model which imitates the target model. However, while knowledge distillation environments typically retain full knowledge of the target model training data and weights, black-box model stealing eliminates the need to have access to the target model weights and training data. Going one step further, black-box model stealing typically uses real-world data samples to train the student network, whereas data-free model stealing removes that requirement by leveraging a generator. As the relationship between the substitute dataset and true dataset is typically unknown, the removal of the substitute data results in more generalizable approaches.

Existing methods for data-free model stealing involve either using a min-max adversarial approach (Truong et al., 2021; Kariyappa et al., 2021; Zhang et al., 2022), or a generator-discriminator setup incorporating Generative Adversarial Networks (GANs) (Sanyal et al., 2022). DFME (Truong et al., 2021) and DFMS-HL (Sanyal et al., 2022) are the two most recent and highest performing methods for each of these approaches, respectively. In both approaches, a student is optimized to minimize the distance between the student and target model outputs. However, in the former, a generator is optimized to maximize the distance between student and target model outputs, while in the latter, the generator-discriminator is optimized to generate samples similar to a synthesized dataset which have balanced student classifications. Even though utilizing a synthesized dataset relaxes the truly data-free requirement, the current state of the art (SOTA) is provided by the DFMS-HL method which *does* utilize synthetic data, while DFME method provided the previous SOTA and *doesn't* require synthetic data.

Our proposed Dual Student method alters the generator-student min-max framework by training two student models to match the target model instead of one. This allows the generator to use one of the students as a proxy for the target model, resulting in a much more effective training criterion. The two students are symmetrically trained to match the target model, while employing the new generator objective to generate samples on which the two students disagree. Disagreement between the two students implies disagreement between at least one of the students and the target model, resulting in the generated sample being a hard sample (c.f. hard sample mining (Shrivastava et al., 2016)) to at least one student. This hard sample provides the driving force to explore controversial regions in the input space on which the students and teacher disagree on. Finding samples on which the student and teacher disagree is also an implicit goal of DFME. However, while DFME does optimize the generator to find disagreement between the student and target, it uses gradient estimations of the target. Our method, on the other hand, uses the true gradients of the second student which can be obtained using direct backpropagation.

Using two student models as a proxy for the target model results in better target model gradient estimates than data-free model stealing methods like DFME, which use explicit gradient estimates. Formalizing our setup, substituting a second student model for the target model creates an alternative lower bound for the student-teacher optimization. Changing the training objective of the generator-student framework to more directly optimize to match the target model results in student model gradients being better aligned with the target model gradients. In our experiments, we show the gradients used when training dual students is closer to using the true gradients of the target model than the estimation method used by DFME (Truong et al., 2021). This important change eliminates the need of some existing approaches (Truong et al., 2021; Kariyappa et al., 2021) to estimate the gradient through the target model. Gradient estimation techniques used in these existing works assume the target model provides soft-label outputs - removing the gradient estimation removes this dependence on soft-labels and allows existing methods to be easily extended to the more difficult hard-label setting. Removing the gradient estimation also reduces the number of queries made to the target model. Finally, we also explore a more difficult data-free setting where the number of classes is not know in advance. To the best of our knowledge, we are the first to address this setting and provide a solution to this new challenging data-free setup.

In summary, our contributions in this paper are as follows: (1) We propose the Dual Student method which provides a tractable approach for the data-free model extraction problem with minimal added components. This includes a mathematical reframing of the data-free model stealing objective along with an empirical comparison with the previous formulation provided by DFME (Truong et al., 2021). (2) The Dual Student method can be incorporated into existing soft-label approaches to extend them to hard-label settings, and has better classification accuracy than existing methods. (3) We show the effectiveness of utilizing the Dual Student setup to fine-tune fully trained models by

directly improving the existing SOTA from DFMS-HL and evaluate the effectiveness of the top data-free model stealing approaches for transfer-based adversarial attacks against the target model.

## 2  RELATED WORK

In this section, we outline the foundational papers our method is built upon, as well as other recent works in this area. We start with knowledge distillation - distilling the knowledge out of a target model and into a student. Then we describe the transition towards model extraction and model stealing, and its extension into the data-free model domain. Finally we discuss the use of data-free model stealing for transfer-based adversarial attacks.

### 2.1  KNOWLEDGE DISTILLATION

Knowledge distillation is the process of using transfer learning to train a different, typically smaller, model. From the area of model compression (Buciluǎ et al., 2006; Romero et al., 2014), knowledge distillation was initially used to determine the necessary depth of deep learning models in Ba & Caruana (2014). The objective was then expanded towards the general goal of creating smaller and more efficient models (Hinton et al., 2015; Urban et al., 2016; Yin et al., 2020). Our approach to data-free model extraction builds off of knowledge distillation techniques for training a student model from a target model, otherwise referred to as the "teacher" or "victim" model in other methods.

Model extraction, also called model stealing, is used to obtain a new model which performs similarly to the target model without knowledge of the target model's gradients. First uses of model extraction were in extracting decision trees to help better explain deep learning models (Bastani et al., 2017b;a), however, recent applications have focused on obtaining a substitute model for the purpose of transfer-based attacks on the target model (Wang et al., 2021). Though early knowledge distillation works relied on access to the original training data, more recent data-free knowledge distillation methods utilize a generator to remove this dependence (Fang et al., 2019; Binici et al., 2022). Recent works utilize activation regularization between the target and student models, in addition to interpolation between generated examples to improve training (Qu et al., 2021).

### 2.2  DATA-FREE MODEL EXTRACTION

Data-free model extraction aims to train a student model without being given a dataset prior. Most existing model extraction techniques rely on either some small set of labeled samples, or an alternative training dataset (Orekondy et al., 2019; Barbalau et al., 2020; Wang, 2021). The first solutions to this data-free model extraction problem are called DFME and MAZE, and use an adversarial setup between a generator and a student (Truong et al., 2021; Kariyappa et al., 2021). The student aims to match the target model and the generator aims to find examples which result in different classifications between the student and target models. However, these methods rely on estimating the gradient of the target model, so more recent work has focused on removing this dependence by altering the generator objective (Hong et al., 2022; Sanyal et al., 2022). The current SOTA for data-free model stealing is DFMS-HL by Sanyal et al. (2022), which not only sets the bar for student model classification accuracy, it also extends the problem to the hard-label setting. Their approach utilizes a full generator-discriminator GAN setup to generate samples similar to a synthetic dataset while retaining balanced class classifications of the samples by the student model. These two aspects – keeping samples similar to the synthetic dataset, and balancing the generated class distribution – remove the reliance on gradient estimation techniques that methods such as DFME require. We note that DFMS-HL utilizes a synthetic dataset of random shapes generated on colored backgrounds, which breaks the truly data-free setting.

### 2.3  TRANSFER-BASED ATTACKS

Adversarial attacks (Szegedy et al., 2013; Akhtar & Mian, 2018; Kardan & Stanley, 2017; Kardan et al., 2021; Beetham et al., 2022) pose a significant obstacle against deploying deep models. In particular, transfer-based adversarial attacks Akhtar et al. (2021) use a substitute model's gradients to generate an adversarial image to fool the target model. This setting was explored extensively by early methods, however, recent work has moved towards a more realistic data-free setting (Zhang

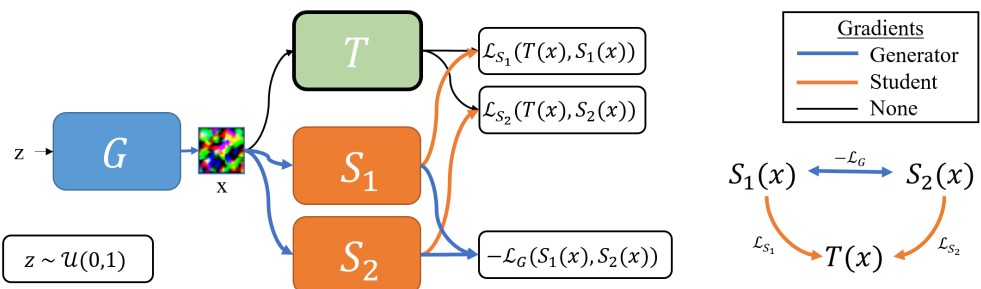

Figure 1: Our network consists of a generator, two students, and a target model. The generator generates images which the two students are trained to classify. The student losses in the Dual Student setup aim to reduce the distance between the target and student model outputs, and only require the soft-label or hard-label outputs of the target model. The generator loss competes against the two student losses by maximizing the distance between the target and student model outputs.

et al., 2021). A prominent use-case for model stealing attacks is to use the student model as a proxy for transfer-based adversarial attacks (Hong et al., 2022; Li & Merkel, 2021; Yue et al., 2021). The method proposed in Zhang et al. (2022) leverages data-free model stealing work to use fewer queries to train a substitute model for transfer-based adversarial attacks. However, we do not include this work in our transfer-based attack comparison, as that work is targeted towards transfer-based adversarial attacks as opposed to our primary objective of mimicking the performance of a classification model.

## 3 DUAL STUDENTS METHOD

This section outlines the framework of the Dual Students method. We first explain why our proposed optimization is better, and then provide an empirical analysis of our algorithm. The Dual Students method consists of a generator, two students, and a target model with limited black-box access. Similar to the adversarial objective in GANs where the generator tries to fool the discriminator, the generator in the Dual Students method generates images on which the two students should have disagreeing outputs. A mismatch between the outputs of the two students on a generated image implies that at least one of the student outputs does not match the teacher output for this image. This is beneficial for optimizing the students to match the target model, especially in the hard-label setting where hard examples are crucial for correcting the student models. These differing objectives – generator promoting disagreement between the students and the students aiming for agreement with the teacher – form the min-max framework of our setup, which facilitates the exact computation of the gradients for all losses.

Architecture of the proposed method is show in Figure 1, where a key contribution is that our generator optimization no longer requires the estimated gradient of the target model. In our method, the student model gradients are not used as a direct proxy for the target model gradients. Despite this, when compared with ideal gradients from a student model trained in a white-box setting, our method results in student-model gradients closer to the white-box setting than the single-step forward-differences method.

### 3.1 DATA-FREE MODEL STEALING FRAMEWORK

For target $T$, generator $G$, student $S$, and noise $z$, our broad objective for solving data-free model stealing is to optimize:

$$\min_S \max_G ||T(G(z)) - S(G(z))||_p. \tag{1}$$

The original Data-Free Model Extraction framework aimed to extract the target model by utilizing a generator $G$ with parameters $\theta_G$ to generate images $x$ from noise $z \sim \mathcal{U}([0, 1]^d)$, where $d$ is the dimension of the latent space, in which the student $S$ with parameters $\theta_S$ is trained to match the

output of target model $T$. The loss function for the student is:

$$\min_{\theta_S} \mathcal{L}_S(S(G(z;\theta_G);\theta_S), T(G(z;\theta_G))),$$

where student loss $\mathcal{L}_S$ is typically $\mathcal{L}_S = D_{KL}$ or $\mathcal{L}_S = \ell_1$. We choose to use the latter when applicable, as it has been shown to perform better than $D_{KL}$ (Truong et al., 2021). The generator, on the other hand, is optimized to maximize the difference between student and teacher. To this end, the negative of the student loss is typically used:

$$\min_{\theta_G} -\mathcal{L}_G(S(G(z;\theta_G);\theta_S), T(G(z;\theta_G))).$$

However, whereas the student loss only requires the gradient through the student model, the generator loss requires the gradient through the student, target, and generator. Due to the limited black-box access to the target model, existing approaches utilize the forward differences method to estimate the gradient through the target model (Truong et al., 2021). An additional problem is that the generator loss does not directly promote a diversity of classes within the generated images. Though this has not been a problem for 10-class datasets like SVHN and CIFAR10, we note in Section 4.2 that diversity of classes becomes a larger challenge when using datasets with more classes like CIFAR100.

## 3.2 Approximating Gradient with Dual Students

Towards making the broad DFMS objective outlined in Eq. 1 differentiable, we propose adding an additional student $S_2$ to solve the following optimization problem:

$$\min_{S_1, S_2} \max_G ||S_1(x) - T(x)||_p + ||S_2(x) - T(x)||_p, \tag{2}$$

for any $\ell_p$ norm loss. Though, neither the original objective in Eq. 1 or this new objective are differentiable due to requiring the gradients through the Target model $T$, adding the second student $S_2$ allows us to make a substitution to use the generator optimization:

$$\max_G ||S_1(G(z)) - S_2(G(z))||_p. \tag{3}$$

This optimization for Generator $G$ removes the Target model $T$ and makes the problem directly differentiable. This new generator optimization is related to our new objective in Eq. 2 through the triangle inequality,

$$||S_1(x) - S_2(x)||_p \le ||S_1(x) - T(x)||_p + ||S_2(x) - T(x)||_p. \tag{4}$$

The LHS of the inequality in Eq. 4 corresponds to the generator's loss in Dual Student with $x = G(z)$ in Eq. 3, and the RHS is the minimization desired in Eq. 1 with an additional student. In other words, when the LHS is maximized during generator optimization, the lower bound is increasing:

$$\max_x ||S_1(x) - S_2(x)||_p \le \max_x ||S_1(x) - T(x)||_p + ||S_2(x) - T(x)||_p.$$

This ensures that either $S_1(x)$ or $S_2(x)$ is getting farther away from $T(x)$ during the generator optimization. To optimize min-max optimization in Eq. 3, we iteratively update the student and generator networks. In the first step, Student $S_i$ with parameters $\theta_{S_i}$ is optimized to match the output of the target model for a particular query:

$$\min_{\theta_{S_i}} \mathcal{L}_S(S_i(G(z;\theta_G);\theta_{S_i}), T(G(z;\theta_G))).$$

Next, the generator is updated according to:

$$\max_{\theta_G} \mathcal{L}_G(S_1(G(z;\theta_G);\theta_{S_1}), S_2(G(z;\theta_G);\theta_{S_2})). \tag{5}$$

The student loss used for soft-labels is $\mathcal{L}_S = \ell_1$, whereas the student loss used for hard-labels is cross entropy, i.e. $\mathcal{L}_S = ce$, as this is standard for classification tasks. The generator optimization maximizes the difference between the outputs of the two students, and $\mathcal{L}_G = \ell_1$ is used in the generator loss when training on both soft-labels and hard-labels. We provide an algorithm for our method in the supplemental materials.

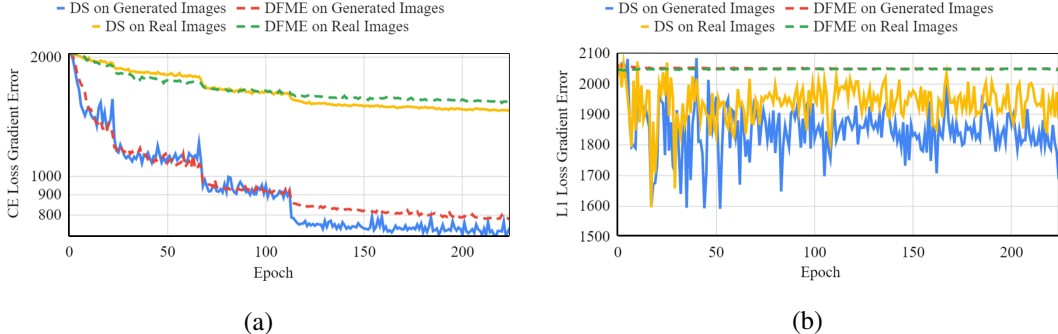

(a)                                           (b)

Figure 2: The distance between the true and estimated target model gradients of either Generated images or Real images. The gradient is computed w.r.t. input image $x$, and a formal description of the distance used is provided in Eq. 6. (a) Cross-Entropy loss is used to compute the gradient in the hard-label setting, and (b) $\ell_1$ loss is used in the soft-label setting. Real images are from the test split of the CIFAR10 dataset.

A major benefit of our approach is the simplicity of requiring only an additional student to overcome the non-differentiability of the black-box teacher. Our method is fully differentiable, in contrast to methods like DFME which approximate teacher gradients using forward differences. Additionally, the Dual Students approach is compatible with both soft-label and hard-label learning through a simple change to the loss function. Existing gradient-estimation based methods like DFME can only be used in the soft-label setting due to the forward differences method needing the additional information provided by the soft-label to remain effective. For other fully trained DFMS, our method can be used to fine-tune the trained models as long as there is a student and generator. In the next subsection, we evaluate the accuracy of the different gradient approximation methods compared to our Dual Students method.

### 3.3 EVALUATING THE GRADIENT APPROXIMATION

The aim for the generator optimization under the black-box scenario proposed in Eq. 5 is to get as close as possible to the generator optimization under the white-box scenario outlined in Eq. 1. One way to evaluate the quality of the estimated gradients between the forward differences method and the Dual Students method is to compare the gradients computed. Specifically, the gradients from $\mathcal{L}_G$ w.r.t. the generated image $x$ when using the black-box estimate, versus the gradients computed when using the white-box Target model directly. By comparing difference between the normalized gradients, the difference can be shown as:

$$f(x) = \left|\left|\nabla_x \frac{\mathcal{L}_G(S(x;\theta_S), T(x;\theta_T))}{||\mathcal{L}_G(S(x;\theta_S), T(x;\theta_T))||} - \nabla_x \frac{\mathcal{L}_G(S_1(x,\theta_{S_1}), S_2(x;\theta_{S_2}))}{||\mathcal{L}_G(S_1(x;\theta_{S_1}), S_2(x;\theta_{S_2}))||}\right|\right|_2 . \quad (6)$$

The results of the forward differences method compared with our Dual Students using this setup is shown in Figure 2. The figure plots the difference between the normalized gradient of the generated image using the target model (white-box gradient) versus the estimate (forward differences or Dual Students). Figure 2a shows that our method provides a gradient estimate closer to the white-box setting than the forward differences method used in DFME in the hard-label setting for both generated and real images. Figure 2b shows error for the soft-label setting, where we see DFME has similar estimation error between real and generated images, while our Dual Students has a smaller error for both between real and generated images. We note that the performance on generated images are more important as generated images are the only images the models have access to during training.

Empirically, we see that this new generator loss results in more accurate gradient estimations for the target model than the forward differences method, as is shown in Figure 2. We note that the forward differences method utilizes soft-labels, and performance is much worse when the soft-label information is reduced to hard-labels.

Table 1: Percent student accuracy for different methods with and without the Dual Student (DS) method.

| Dataset | Target Accuracy | Method | Probabilities | Hard-Labels |
|---------|----------------|--------|---------------|-------------|
| MNIST | 99.66 | DFME | 99.15 | 97.85 |
| | 99.66 | DS | **99.36** | **99.25** |
| FashionMNIST | 93.84 | DFME | 85.17 | 48.91 |
| | 93.84 | DS | **91.17** | **80.53** |
| GTSRB | 97.21 | DFME | 96.35 | 85.69 |
| | 97.21 | DS | **96.40** | **93.20** |
| SVHN | 96.20 | DFME | 95.33 | 93.87 |
| | 96.20 | DS | **95.72** | **95.43** |
| CIFAR10 | 95.5 | DFME | 88.10 | 68.40 |
| | 95.5 | DS | **91.34** | 78.72 |
| | 95.5 | DFMS-SL/HL | 88.51 | 79.61 |
| | 95.5 | DFMS-SL/HL + DS | 89.38 | **85.06** |
| CIFAR100 | 77.99 | DFME | 26.46 | 6.91 |
| | 77.99 | DS | 45.32 | 9.77 |
| | 77.99 | DFMS-SL/HL | 44.86 | 35.78 |
| | 77.99 | DFMS-SL/HL + DS | **50.98** | **36.38** |

# 4 EXPERIMENTS

We evaluate the Dual Students method on various datasets for standard classification accuracy and transfer-based attack effectiveness. The datasets used are MNIST, FashionMNIST, GTSRB, SVHN, CIFAR10, and CIFAR100 (Xiao et al., 2017; Stallkamp et al., 2011; Netzer et al., 2011; Krizhevsky et al., 2009). The target model architecture is ResNet-34, while the students are ResNet-18 and are trained for 2 million queries for MNIST, FashionMNIST, GTSRB, and SVHN and 20 million queries for CIFAR10 and CIFAR100 (He et al., 2016). The generator architecture is the same 3-layer convolutional model used in DFME. For the classification task an average of the two student's outputs are used in an ensemble. However, for transfer-based attacks, only the higher accuracy student is used as a proxy in order to help make the performance more comparable across methods. More experiment details are provided in the supplemental material.

## 4.1 CLASSIFICATION PERFORMANCE

The standard evaluation setting for data-free model extraction is to test the student model accuracy on the test set respective to the dataset the target model was trained on. Table 1 provides a comparison between DFME (Truong et al., 2021), DFMS-HL (Sanyal et al., 2022), our proposed Dual Students method, and when our Dual Students method is used with DFMS-HL. As can be seen, our method outperforms DFME and DFMS-HL on the different datasets using soft-labels (probabilities), with the most notable increase being on FashionMNIST. On hard-labels, our method outperforms DFMS-HL when used to fine-tune the DFMS-HL trained student.

Using Dual Students to fine-tune fully trained generator-student-teacher methods is an effective way to increase accuracy. Specifically, we take the fully trained student from one method and insert it as a pretrained method into our Dual Students setup. At the start of fine-tuning using Dual Students, the generator along with one of the students is uninitialized, while the remaining student uses the weights trained by the original method. After fine-tuning, only the fine-tuned student is used during evaluation. This fine-tuning is especially beneficial for methods that have a different optimization scheme such as DFMS-HL. From a high level view, the DFMS-HL method optimizes the generator to generate images that are similar to the proxy data and that have a balanced distribution of classes, when classified by the *student model*. Due to the student model being optimized to match the target model outputs, the generator is indirectly optimized to generate a balanced distributions of classes as classified by the *target model*. There is no loss in the setup which directly incentivizes samples on which the student and teacher disagree. Thus when our method is used to fine-tune the fully trained DFMS-HL models for 1 million queries (5% of total), we see an improvement of over 5% in Table 1. We note that the DFMS-SL/HL results shown were trained using our target model in order to better

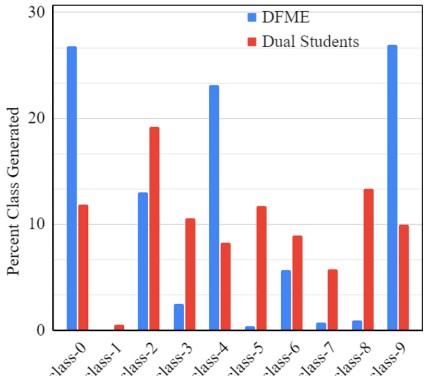

Figure 3: Class distribution of generated classes at the end of training for CIFAR10 dataset. Dual Student is able to produce more balanced samples.

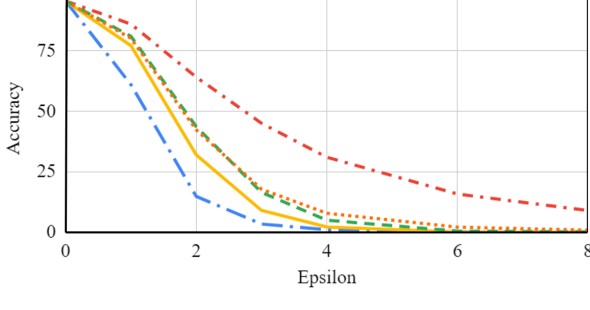

Figure 4: Accuracy on Target Model of perturbed images generated using PGD attack with varying epsilons ($\frac{\epsilon}{255}$) where different soft-label DFMS student models are used.

compare with other methods, and don't fully match the accuracies reported in Sanyal et al. (2022). They achieve 85.92% accuracy on hard-labels compared with the 79.61% accuracy we found. When their model is fine-tuned using Dual Students, we observe an increased accuracy of 88.46%.

## 4.2 QUERY EFFICIENCY AND NUMBER OF CLASSES

The Dual Student generator loss avoids the extra queries necessary for zeroth order gradient estimation when optimizing the Generator. Using zeroth order gradient estimation adds additional queries to the target model during the generator training. The DFME method has the same alternating generator-student training and uses the same parameters generator and student iterations. However, due to their use of the forward differences method when updating the generator, each generator iteration requires an additional 2 queries to the target model. This results in our method using $\frac{5}{7} \approx 71.42\%$ of the queries that DFME does.

When compared with the DFMS-HL, our Dual Student method benefits from a simpler training setup and requires less compute. Though both methods have the same number of separate models, DFMS-HL requires the generator to be pretrained on the synthetic data, whereas our method can be trained from scratch using alternating generator-student training loops.

DFMS faces challenges when scaling to datasets with larger numbers of classes. Intuitively, datasets with more classes tend to be more difficult than datasets with fewer classes, and this is what we found when extending our method to CIFAR100. With no changes to our setup, DFME achieves around 26.46% accuracy on CIFAR100, while our Dual Students method achieves around 45.32%. A reason for this large reduction in accuracy, when going from CIFAR10 to CIFAR100, is the lack of explicit diversity within the generator loss optimization $L_G$ that we define. Though the generator is optimized to find confusing images for the two students, there is no guarantee that those confusing images are evenly distributed across classes. Despite this, Figure 3 shows that the generator for the Dual Students method generates a more balanced set of classes than the DFME generator on CIFAR10, where perfectly balanced classes would be 10% prevalence for all classes. We provide additional figures and analysis of generated image class distributions in our supplementary material.

A natural extension to the data-free black-box hard-label environment that DFMS works in is to remove the knowledge about the number of classes in the target dataset. This raises the challenge of how to learn about the number of classes, and how to alter the classifier to grow as the number of classes grows. To extend our method and address these challenges, the seen classes are mapped in the order they are observed, and student classifier training is limited to the number of seen classes. As additional classes are discovered naturally by the generator traversing the image-space, the final classification layer of the student model expands to include these additional classes as separate outputs. Training on CIFAR10 using hard-labels without the number of classes known, our method achieves 74.95% accuracy, which is comparable to the hard-label accuracy when the number of classes are known.

Table 2: Attack success percentage of different DFMS methods on the Target Model trained on CIFAR10 when attack $\epsilon = \frac{3}{255}$. All attacks are evaluated on the Target Model. The Target Model row is a white-box attack, Proxy Model is a transfer-based black-box attack where the proxy is trained using the same data as the target model. The other DFMS methods provide trained student models which act as the proxy in transfer-based black-box attacks.

| Attack | Method | Untargeted Attacks | | Targeted Attacks | |
|---|---|---|---|---|---|
| | | **Probabilities** | **Hard-Labels** | **Probabilities** | **Hard-Labels** |
| FGSM | *Target Model* | 45.00 | | 19.64 | |
| | *Proxy Model* | 33.12 | | 14.38 | |
| | DFME | 56.84 | 39.22 | 21.07 | 17.15 |
| | DS | **62.35** | 44.58 | **21.58** | 21.04 |
| | DFMS-HL/SL | 54.88 | 48.89 | 19.74 | 21.85 |
| | DFMS-HL/SL + DS | 54.99 | **50.41** | 20.53 | **23.59** |
| PGD | *Target Model* | 96.78 | | 76.32 | |
| | *Proxy Model* | 55.01 | | 28.33 | |
| | DFME | 83.59 | 54.71 | 51.97 | 31.49 |
| | DS | **91.04** | 62.21 | **61.96** | 33.39 |
| | DFMS-HL/SL | 81.97 | 72.40 | 52.64 | 39.95 |
| | DFMS-HL/SL + DS | 81.04 | **73.08** | 51.38 | **42.53** |

## 4.3 BLACK-BOX ATTACKS

The effectiveness of using the trained student model as a substitute for the target model in a black-box transfer-based adversarial attack can be seen in Table 2. The DaST method (Zhou et al., 2020) and the work done by Zhang et al. (2022) also have a similar generator-student setup, however the focus of those works are primarily on being used as a transfer-based proxy model. Our work, on the other hand, focuses on the classification accuracy of CIFAR10. In the fooling rates in Table 2, we can see that our method has a much higher fooling rate for the different attacks. We chose to use a small epsilon of $\epsilon = \frac{3}{255}$ instead of the standard $\epsilon = \frac{8}{255}$ to highlight the difference in effectiveness. For higher epsilons the attack success rate becomes 100%, as can be seen in Figure 4.

Two observations of note in Table 2 are that the data-free methods have a higher attack success rate than the proxy model trained on the real dataset, and that the data-free methods outperform the white-box attack when using FGSM (Goodfellow et al., 2015). For the former, it might seem counter intuitive that the student models significantly outperform the regularly trained proxy model. However, this result is expected, as the millions of queries to the target model used to train the student model result in the students having gradients more similar to the target model. For the latter, the DFMS methods having a higher fooling rate than the white-box setting is surprising. One explanation for this is that the FGSM attack is more susceptible to noise within the attacked-model's gradients due to only taking a single step in the direction of the signed gradient. The white-box model is a larger model trained on the real dataset, and thus has a higher tendency for noise within its gradients as compared with the student models. The student models, on the other hand, are trained on images designed to be near the class boundaries, as, if there is noise within the student model gradients, the generator will leverage that noise to cause disagreement between the students. PGD takes multiple smaller steps and is thus less susceptible to the noise which explains why the white-box setting is more optimal that attack (Madry et al., 2017).

## 5 CONCLUSION

In summary, we outlined a new Data-Free Model Stealing optimization cultivated in the Dual Students method. We showed the Dual Students method is effective in stealing the functionality of the target model as well as providing a good proxy from which to generate transfer-based adversarial attacks. Towards improving model stealing methods, reducing the number of queries required and extending to more complex datasets and tasks are major challenges to be explored in future work.

**Ethics Statement:** Advancing the model stealing area has the potential to improve the effectiveness of attacks from adversaries with ill intentions. However, for the data-free area specifically, this is not an immediate concern due to current DFMS methods being very limited when it comes to scaling to more useful or valuable problems. And, as with many areas of machine learning research, it is not clear that work in this area will not bring about benefits towards knowledge distillation and machine learning as a whole which may outweigh potentially exacerbating the model stealing threat.

**Reproducibility Statement:** We outline the general idea of our method and provide the key hyperparameters used during training. In the supplemental material we provide the algorithm as well as additional hyperparameters necessary for replicating the results shown in the tables. Because the dual students method does not require pretraining, other synthetic datasets, or models that require separate training routines, the environment can be setup and trained from scratch without using proprietary datasets or environments.

## 6 ACKNOWLEDGEMENTS

We thank Dr. Mitch Hill for insightful discussions and feedback. This material is partially based upon work supported by the Defense Advanced Research Projects Agency (DARPA) under Agreement No. HR00112090137, and is approved for public release; distribution is unlimited. Professor Ajmal Mian is the recipient of an Australian Research Council Future Fellowship Award (project number FT210100268) funded by the Australian Government.

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
