# OpenReview forum: "Dual Student Networks for Data-Free Model Stealing"
_ICLR.cc/2023/Conference — ICLR 2023 poster_

### Official Review · Reviewer_aGyH · 2022-10-21

**Confidence:** 4
**Correctness:** 4
**Technical Novelty And Significance:** 4
**Empirical Novelty And Significance:** 4
**Recommendation:** 8

**Clarity, Quality, Novelty And Reproducibility:**

The paper is clear. The contributions are well motivated and fairly new.
The authors provide a reproducibility guidelines in the paper.

**Strength And Weaknesses:**

Pros :
- the paper is well written and pleasant to read
- the proposed contribution requires fewer queries to the to-be-stolen teacher model and is also applicable in situations where the number of classes is not known in advance.
- the proposed approach is fully differentiable which is a nice increment over SOTA.

Cons :
- the paper does not mention computation time/complexity overhead incurred by using an additional student

**Summary Of The Paper:**

This paper adresses data-free model stealing, an ambitious framework where one attempts to build a model that imitates a target one without accessing the target parameter or any training data point (true or substitute ones). In this field, SOTA approaches use a student network whose outputs are meant to minimize a distance between its outputs and the outputs of the target (which is the only information unveiled by the target model). The student competes with a an adversary as part of a min-max game in order to efficiently explore the input space.

Compared to previous approaches, the authors use two students and exploit disagreement between the two to better explore the input space in regions where harder inputs live. Disagreement is exploited by an adversary input generator which will try to maximize conflict. The generated inputs are the one on which student and teacher prediction should align.

**Summary Of The Review:**

Major remarks:

The only missing aspect in the paper is an analysis of increased time/memory requirements of the method  compared to prior arts.

Let alone that, the paper is interesting and I do not have compelling remarks to address but rather a few questions that came to my mind :

Is there any benefits in training more than 2 students ? Perhaps more precise finer-grained disagreement regions could be obtained in this way.

Do the author have an opinion on possible counter-measures that a defender could implement to misguide the students ?

Minor :

In some parts of the paper, the authors use the word « image » as if it was obvious that inputs are images.

---

> ### Author Response · Authors · 2022-11-10
> **Responses to questions along with notes of changes made to the paper**
>
> Thank you for your review of our paper. We’ve made the minor requested changes and have provided a more detailed breakdown of the compute differences between the two most comparable methods (DFME and DS). Please let us know what additional changes we can make to strengthen our paper and if you have any other questions or concerns.
>
> 1. Although we don’t provide an explicit analysis of the compute, we do make a note of it in section 4.2 paragraph 2. In general, the key criteria data-free model stealing looks at is query efficiency and classification accuracy. Compared with DFME, our DS method has an extra student to train and thus requires additional compute. However, DFMS-SL/HL requires more compute than our method despite having the same number of models (student-generator-discriminator vs student-student-generator) due to the pretraining required for the generator. That being said, the best performing method is a student trained by DFMS-HL then fine-tuned using DS, which requires the most compute of the evaluated methods.
>
> 	To be more precise, with a batch size of 256 and a 20 million query budget (i.e. 20m/256  target model forward passes), DFME training on CIFAR10 requires (20m/256/7=) 11k generator backward passes and (20m/256/7\*6=) 67k generator forward passes while DS training requires (20m/256/5=) 16k generator backward passes and (20m/256/5\*6=) 94k forward passes. For student training, DFME uses (20m/256/7\*6=) 67k student backward and forward passes, while DS uses (20m/256/5\*6\*2=) 188k backward and forward passes for both students combined. The increase in compute comes from better use of target model queries – the gradient estimation in DFME is not required for DS – and the addition of the second student.
>
> Response to Summary.
>
> 1. S) Anecdotally yes, there appears to be some minor benefits to adding an additional student. However, adding additional students requires changes to the loss in order to better promote class diversity. At a high level there’s reason to think adding additional students may help or harm training.  On one hand, adding additional students can be viewed as pushing the generator in a more average direction – the more students, the more stable the generator optimization. However, this may be detrimental to training because some instability may be beneficial for the generator in exploring the image space. On the other hand, adding additional students may promote the generator exploring more diverse spaces by being more in-line with the target model. The only definite detriment to adding additional students is the added computation time. Of note is that the Dual Student setup would not require additional queries if the number of students increases. This is an interesting question and may warrant further study.
>
> 2. S) As data-free model stealing tends to be weaker than traditional model stealing, it follows that model stealing defenses should also work on data-free model stealing. Additionally, due to the generated images being far from the real image space, an out of distribution detector could trivially detect samples generated using data-free methods.
>
> 3. S) We have changed “image” to more general terms in the abstract and introduction to keep things general initially, and only use “image” after Figure 1 and the introduction to our method specifically.

---

### Official Review · Reviewer_yxtQ · 2022-10-22

**Confidence:** 3
**Correctness:** 2
**Technical Novelty And Significance:** 2
**Empirical Novelty And Significance:** Not applicable
**Recommendation:** 3

**Clarity, Quality, Novelty And Reproducibility:**

The paper is reasonably clear. It is well written in good English and the reported results are encouraging. The authors haven't provided the code, however I expect it will be available at some point facilitating the reproducibility of the method. While the model is interesting, certain design choices are not clear or well ablated and some important experiments (CIFAR 100) are missing.

**Strength And Weaknesses:**

**Strengths**

(1) The paper addresses an interesting problem of data-free learning or data-free model stealing, where they proposed a dual-student based framework.

(2) The paper has reported encouraging results on the available benchmarks for both standard classification accuracy and transfer-based attack effectiveness.

**Weaknesses**

(1) It is not clear why a dual student model is needed for this. How are they creating more effective criteria? In my understanding it can also be done with the target model and only one student. In that case, the only student will be trained to match the target model, while the generator can be tasked to generate samples on which the student and the target model disagree for creating hard samples. More details with writing is needed together with appropriate ablation study.

(2) There are some clarifications on the reasons at the very last paragraph of section 3.1, however those reasoning aren't well grounded. So please cite the papers that utilize the forward differences method to estimate the gradient through the target model. Additionally, ablative studies should be performed or indicated to justify the design choice.

(3) Results on CIFAR 100 dataset are missing in table 1. As the model is motivated based on its effectiveness of generating more diverse samples to be effective for datasets with more number of classes, results on CIFAR 100 is important. In table 1 of the supplementary material, both the rows show results on CIFAR 10, is it correct? If so, why are the numbers in the DFMS-HL/SL rows different?

**Summary Of The Paper:**

This paper has addressed the problem of data free model stealing, where a dual student based framework is proposed for better estimating gradients of the target model without access to its parameters, and generating a diverse set of images that thoroughly explores the input space. While the proposed framework looks interesting, its design is not well justified and not well ablated.

**Summary Of The Review:**

The paper is interesting with a new architecture, but the design choices are not well grounded. Some important experiments are missing which makes it difficult to judge the effectiveness of the proposals.

---

> ### Author Response · Authors · 2022-11-10
> **Responses to questions along with notes of changes made to the paper**
>
> Thank you for your review of our paper. We have made the requested changes and reworked parts of the paper to help improve clarity. Of particular note, we hope our response to weakness 1 provides a convincing motivation for why this work is worthwhile. Please let us know what additional changes we can make to strengthen our paper and if you have any other questions or concerns.
>
> 1. We outline our motivations for using two students in equations 1 through 6 – we add an additional student to create a more direct optimization for training the student to match the target. As you describe, there are works like DFME that use a single student and target (and generator), however these methods require a gradient estimation through the target model in order to train the generator whereas our method does not (Figure 2 shows that our gradient estimation is more accurate). More recent works like DFMS-SL/HL use an additional discriminator in order to train the generator in addition to the student and target models. When extending model stealing to the data-free domain, generating good training images becomes a very challenging problem, as initially there is no knowledge about the image space. In non-data-free areas, a single student-target-generator setup is common because they can leverage real-world data into their generator training.
>
> 	What kind of ablation would you like to see? We provide an analysis of the quality of Dual Student estimated gradients in Figure 2 as well as a comparison with and without the Dual Students in Table 1 when we compare DFME with DS.
>
> 2. We have added a citation for the last paragraph in section 3.1. From our proposed change in Equations 1 through 6, we support our reasoning with empirical evidence in Figure 2 that our method results in better gradient estimates than the forward differences used in DFME.
>
> 3. Inline with our response to the 4th point made by reviewer xQYc, CIFAR-100 results are limited to in-text reporting in paragraph 3 of section 4.2 because we were unable to replicate DFMS-HL on CIFAR-100.
>
> 	Yes, Table 1 in supplementary material should show CIFAR10 for both rows. The key change is the difference in target model – for most of our CIFAR10 experiments we used a single target model. However the reported DFMS-SL/HL results use different target models. We include them for the sake of completeness, but in general, results should only be compared when the target model is the same (i.e. when the target model has the same accuracy). The architectures of the target models are noted in the first paragraph of section 1 in the supplementary table, as well as in the first paragraph of section 4 in the paper.
>
> Response to Summary. We will release the code upon paper publication.

---

> > ### Author Response · Authors · 2022-12-10
> > **An update regarding CIFAR-100 results**
> >
> > An update to our response to weakness 3: we have obtained additional CIFAR-100 results shown in the table within our response to reviewer xQYc. CIFAR-100 results will be added to Table 1 in the updated version of our paper.

---

### Official Review · Reviewer_xQYc · 2022-10-26

**Confidence:** 3
**Correctness:** 2
**Technical Novelty And Significance:** 3
**Empirical Novelty And Significance:** 2
**Recommendation:** 5

**Clarity, Quality, Novelty And Reproducibility:**

The proposed dual-student method is interesting, but the authors fail in conducting a comprehensive empirical study.

**Strength And Weaknesses:**

The proposed dual-student method is interesting and empirical results show that it surpasses existing data-free approaches in some situations. However, I think the current version of the paper can be improved in many aspects, which are listed below:

1. The writing of the paper is not well, making it a little confusing to follow the ideas behind the paper. Specifically, in Section 1, the authors put most of the details of dual-student in a single paragraph, which makes it difficult for reading. Besides, Section 2 uses too much content in reviewing previous works (more than 1 page), which however is not so necessary for introducing the proposed dual-student methods.

2. The insight of the proposed dual-student method is not clear enough.
    - Why replacing the objective problem in Eq.(1) with Eq.(2) is appropriate? I suggest the author give at least an intuitive explanation about it.
    - An ablation study on the relationship between the step numbers for updating the generator and student networks would help understand why the dual-student method performs well in practice.

3. Some experiment details are missing.
    - What is the model architecture of the query data generator?
    - Please explain how to combine the two methods, DFMS-HL/SL and DS in detail.

4. The authors claim that they conduct experiments on the CIFAR-100 dataset. However, I could not find corresponding experiment results in tables or figures from the paper.

5. Some suggestions about improving the empirical studies.
    - Study the relationship between query budget and stealing performance of the dual-student method.
    - Involve more black-box attacks for evaluation, e.g., membership inference attacks, and model-inversion attacks.

6. Typos: In Eq.(3): $G(x)$ -> $G(z)$.

**Summary Of The Paper:**

This paper studies how to perform model stealing in the data-free setting.
The authors first identify two weaknesses of existing data-free model stealing methods: (1) the generated query data are not informative enough, and (2) the estimation of black-box gradients is not stable for optimization.

To address these two issues, the authors then design a novel dual-student model stealing method.
The dual-student method, which involves two student networks and a query data generator, aims to solve a minimax problem. The two student networks aim to imitate the behavior of the victim model on generated query data distribution, while the generator aims to generate query data that maximize the prediction differences between student networks and the victim.
Through a simple substitution, the solution of the maximization problem can be approximated as maximizing the disagreement between the two student networks, which therefore does not require black-box optimization anymore.

**Summary Of The Review:**

In general, the proposed method is novel and shows some performance advantages. However, the authors did not fully justify its effectiveness. Besides, the writing of this paper needs to be improved. Therefore, I tend to reject this paper for now. I suggest the authors to improve the writing, add more experiments, and submit the paper to another top conference.

---

> ### Author Response · Authors · 2022-11-10
> **Collection of requested results along with explanations and notes of changes made to the paper**
>
> Thank you for your review of our paper. We’ve made the requested major and minor changes – in particular reworking the balance between intro/related works to improve clarity. We hope our explanations resolve the questions raised. Please let us know what additional changes we can make to strengthen our paper and if you have any other questions or concerns.
>
> 1. Splitting the details of DS into 2 paragraphs in the intro and condensing the information in related works is a good suggestion. We have rewritten and expanded parts of the introduction to improve clarity, splitting the problem paragraph into two. We have also cut/condensed information in the related works. As was noted by reviewer vQa1 in their summary, we've gone through and reworked parts to improve the writing.  Most of the paper was noted to be clear by reviewer yxtQ in section “Clarity” and reviewer aGyH in “Pro 1”, but we'd appreciate any additional suggestions to further improve the writing.
>
> 2. a) Conceptually, Eq. 1 optimizes the student s.t. the worst-case generated samples still result in an output similar to the teacher. Eq. 2 is optimizing for the same thing, except adding an additional student: optimize student1 and student2 s.t. the worst-case generated samples result in student outputs similar to the teacher. At a high level, these two equations are optimizing for the same thing – bringing student outputs closer to the teacher outputs. To be more precise, $\min\max 2E\left[|T-S|\right] = \min\max E\left[|T-S_1|\right]+E\left[|T-S_2|\right]$, where expectation is taken over distribution of student models.
>
> 	Our motivation behind making the change from Eq. 1 to Eq. 2 is that Eq. 2 allows us to make a substitution removing the need for directly estimating the target model gradient, which makes the problem more tractable.
>
> 	b) We use a 1:5 ratio between generator iterations and student iterations which follows what the DFME method uses. Below we provide results for the experiments on CIFAR10 soft-labels for DS using other ratios, though there is noise in the final accuracies around the ratio 1:5 - changing to 1:4 or 1:6 does not change the final accuracy much.
> 	| Ratio | 	Accuracy |
> 	|---|---|
> 	| 1:1 | 	66.59 |
> 	| 1:4	| 91.12 |
> 	| 1:5 | 	91.34 |
> 	| 1:6 | 	91.07 |
> 	| 1:10 | 	76.05 |
>
> 3. a) We have added details about the generator architecture to the first paragraph in section 4 - it uses 3 convolutional layers with linear upsampling, batch normalization, and ReLU activations.
>
> 	b) We have changed the 2nd paragraph of section 4.1 to explain this more clearly. The DFMS-HL/SL method fully trains a student (we ignore the generator and discriminator). We then insert that trained student into an untrained DS setup - the DS setup has an untrained generator, a pretrained student1 and an untrained student2. We then proceed with DS training for 1 million queries to fine-tune the pretrained student1, and evaluate only on the fine-tuned student1 (student2 lacks training samples).
>
> 4. The only CIFAR-100 results we provide are in the text as 26.46% accuracy for DFME and 45.32% accuracy using DS, as well as a class distribution graph during training in the supplementary material. We did not include these results in the primary tables as we were unable to replicate the results reported in DFMS-HL. Our reason for including these results is solely for our analysis on the scalability of our method on larger datasets; CIFAR100 results are not a core claim in the paper.
>
> 5. a) We have a graph in the supplementary comparing classification accuracy with query efficiency. We did not include it in the main paper because the results are not straightforward to compare due to the different query efficiencies, the use of moving average in Dual Students, and differing final accuracies.
>
> 	b) The purpose of Table 2 is to compare Attack Success Rate (ASR) within works aiming for best classification accuracy to show the improved internal representations through attack success rate. Methods made specifically for the purpose of attacking can get higher ASR with fewer queries, but they do not retain classification performance. Thus we did not include attack oriented methods because the purpose is to compare between classification methods. Although it would be interesting to compare with more diverse kinds of attacks, that is a bit outside the scope of the project. If there are black-box attacks that fit under the criteria of having decent classification accuracy, we'd be happy to include them as well.
>
> 6. Thank you!

---

> > ### Comment · Reviewer_xQYc · 2022-11-20
> > **Score would not be changed**
> >
> > Thanks to the authors for their response.
> >
> > I appreciate the authors for clarifying the motivation of the dual-student method. Besides, the study on the relationship between generator iterations and student iterations is also interesting, which I suggest including in the main text.
> > Further, additional experiment details (i.e., the generator architecture, and the implementation of DFMS-HL/SL with DS) improve the reproducibility of this work.
> >
> > However, I still think that the proposed method is not well evaluated. Detailed comments are listed below:
> >
> > 1. **Results of CIFAR-100.**
> > I understand sometimes it could be difficult to replicate other people's works. However, as indicated by reviewer yxtQ, results on CIFAR-100 are important since this dataset contains more diverse samples. The authors can report the results of DFMS-HL on CIFAR-100 with their own replicated ones, but missing the performance comparison on CIFAR-100 is inappropriate.
> >
> > 2. **Results of query efficiency.**
> > The authors have indeed reported the query efficiency in supplementary material, but only included **a single** baseline method (DFME), under **a single** experiment setting (soft-labels setting), on **a single** dataset (CIFAR-10). Although the authors argue that it would not be straightforward to present such a query-efficiency comparison due to several reasons, I do not see the difficulties. In fact, the authors could just simply conduct different extraction attacks, record the pairs of query numbers and attack accuracies regularly along the attacks, and finally report the curves of query number-attack accuracy (as that of Fig.1 in the supplementary material). Furthermore, it is worth noting that query efficiency is a crucial metric in the research of model stealing, e.g., Fig.3 in [r1], Fig.3 & Fig.6 in [r2], Fig.5 & Fig.9 & Fig.10 in [r3], and Fig.4 in [r4]. Therefore, I believe including a detailed study on query efficiency is necessary.
> >
> > 3. **Evaluations under more attacks.**
> > Firstly, I could not understand what the sentence *"Methods made specifically for the purpose of attacking can get higher ASR with fewer queries, but they do not retain classification performance"* means. Secondly, here I can explain why I suggest including evaluations under more black-box attacks. As far as I know, a direct application of model-stealing is to obtain a white-box extracted model which can be used as the surrogate model in black-box ML attacks and help improve the performance of these attacks. Similar to the results of adversarial attacks reported in this paper, a better-extracted performance would further indicate a similar performance of other ML attacks between the victim and extracted models. That is why I think these additional metrics based on black-box ML attacks may be important, and I think at least adding evaluation under membership inference attack [r5, r6] would not be a difficult task.
> >
> > For all these reasons, I will keep my score unchanged, although I think the proposed dual-student method is interesting. I suggest the authors perform a more comprehensive empirical analysis and submit this work to another top conference.
> >
> >
> > **References**
> >
> > [r1] Pal et al. "ActiveThief: Model Extraction Using Active Learning and Unannotated Public Data". AAAI 2020.
> >
> > [r2] Truong et al. "Data-Free Model Extraction". CVPR 2021.
> >
> > [r3] Orekondy et al. "Knockoff Nets: Stealing Functionality of Black-Box Models". CVPR 2019.
> >
> > [r4] Oh et al. "Towards Reverse-Engineering Black-Box Neural Networks". ICLR 2018.
> >
> > [r5] Yeom et al. "Privacy risk in machine learn- ing: Analyzing the connection to overfitting". IEEE CSF 2018.
> >
> > [r6] Shokri et al. "Membership inference at- tacks against machine learning models". IEEE SP 2017.

---

> > > ### Author Response · Authors · 2022-12-10
> > > **Requested results, modifications, and explanations**
> > >
> > > Thank you for your updated response.
> > >
> > > 1. **Results on CIFAR-100.** We have modified the publicly available DFMS-HL code to obtain results on CIFAR-100. A few caveats we note are that we do not have the original target model, synthetic dataset, or CIFAR-100 hyperparameters used in [r3]. Instead, we use our own target model and the synthetic dataset/hyperparameters used to train DFMS-HL on CIFAR-10. The results are provided in the table below which we will include in Table 1 in the updated version of our paper.
> > >
> > > 2. **Results on query efficiency.** Query efficiency is a critical metric for this problem, and the first data-free model stealing papers [r1 Fig. 3 Fig. 6,r2 Fig. 3] provided graphs evaluating query efficiency. However, the data-free model stealing methods [r1,r2,r3] all use 20 million queries to get classification results on CIFAR-10, so the more recent work [r3] does not focus on query efficiency outside of an ablation with itself in [r3 Fig. 4]. That said, we agree our supplementary Figure 1 graph is lacking and will add results for DFMS-SL, DFMS-HL, and the hard-label versions of DFME and DS to this figure in our updated version.
> > >
> > > 3. **Evaluations under more attacks.** This is a wide/complex topic - to try to reduce confusion we outline different attack categories and explain why we either do not include these categories, or why we provide the limited examples that we do. Results that we refer to are in Table 2 of our paper.
> > >
> > > 	a) **Apply additional white-box attacks.** We provide results for FGSM, BIM, and PGD, which is in-line with other works that evaluate surrogate model effectiveness in generating transfer-based black-box attacks [r4].
> > >
> > > 	b) **Compare with query-based black-box attacks.** The primary evaluation metric for these attacks is query efficiency instead of attack success rate. Additionally, query-based black-box attacks tend to measure queries per adversarial image generated – whereas we measure the number of queries to fully train a model, whereafter zero queries are required to generate new adversarial images.
> > >
> > > 	c) **Compare with model-stealing black-box attacks.** We do not include work like [r4] because their method focuses on attack success rate and query efficiency whereas ours focuses on classification performance. [r4] appears to sacrifice final classification accuracy for query efficiency, as accuracy results in figures 1,5,6,7 limit the x-axis to a relatively small number of queries. Additionally, the few accuracies reported – [r4 section 4.3] 61.9% on CIFAR-10, [r4 section 4.3] 91.8% MNIST – are far below the classification accuracies of the less query-efficient methods (78.72% and 99.25% on CIFAR-10 and MNIST for DS). Although [r4] has better ASR and uses fewer queries, it is not extracting the target model as closely as standard data-free model stealing techniques. Therefore, including an ASR comparison with [r4] would be misleading.
> > >
> > > 	d) **Compare with membership inference attacks.** It would be interesting to do an in-depth analysis on how well data-free model stealing attacks approximate the target model using a membership inference attack metric instead of the standard classification metric. And that is what we seek to do in our section 4.3 analysis on ASR performance of different data-free model stealing methods. However, membership inference attacks – which seek to determine whether a piece of data was in a target model’s training dataset – does not fit well with the data-free setting, where training data is not available.
> > >
> > > CIFAR-100 Results (Target Model Accuracy: 77.99%):
> > >
> > > | Method | Probabilities | Hard-Labels |
> > > |---|---|---|
> > > | DFME | 26.46 | 6.91 |
> > > | DS | 45.32 | 9.77 |
> > > | DFMS-SL/HL | 44.86 | 35.78 |
> > > | DFMS-SL/HL +  DS | 50.98 | 36.38 |
> > >
> > >
> > > **References**
> > >
> > > [r1] Truong et al. “Data-Free Model Extraction”. CVPR 2021.
> > >
> > > [r2] Kariyappa et al. "MAZE: Data-Free Model Stealing Attack Using Zeroth-Order Gradient Estimation”. CVPR 2021.
> > >
> > > [r3] Sanyal et al. “Towards Data-Free Model Stealing in a Hard Label Setting”. CVPR 2022.
> > >
> > > [r4] Zhang et al. “Towards Efficient Data Free Black-box Adversarial Attack”. CVPR 2022.

---

> > > > ### Comment · Reviewer_xQYc · 2022-12-12
> > > > **Concern about query efficiency**
> > > >
> > > > Thanks to the authors for their response.
> > > >
> > > > I have raised my score to 5, given that the authors provide additional results on CIFAR-100 and carefully explain the relationships between DSMS and other attacks. Please include them in the revised version of the paper.
> > > >
> > > > Now I understand DFMS is more focused on replicating the (classification) ability of target models. However, I still believe it is necessary to compare the query efficiency of DFMS with existing methods:
> > > >
> > > > - Firstly, if an adversary is able to query a victim model for infinite times, he/she can of course replicate the functionality of this victim precisely. However, this is not realistic in real-world scenarios, given that querying MLaaS systems is time-consuming, and there also exist approaches for detecting malicious queries. As a result, query efficiency is an important metric in the context of model stealing.
> > > >
> > > > - Then, regarding this paper, my concern is whether or not DFMS could really achieve better classification performance while maintaining smaller query budgets, compared with previous data-free model stealing approaches. If DFMS requires more queries, then there would be no suprising that it can achieve better classification performance.
> > > >
> > > > I will further raise my score if the authors can address my concern about query efficiency.

---

> > > > > ### Author Response · Authors · 2022-12-12
> > > > > **Response to query efficiency concern**
> > > > >
> > > > > Thank you for your timely response.
> > > > >
> > > > > Regarding query efficiency comparisons, our Table 1 results show that Dual Student classification performance is better when given the **same** query budget. In other words, our method is more query efficient when evaluated at that particular number of queries: 2 million queries for MNIST, FashionMNIST, GTSRB, and SVHN and 20 million queries for CIFAR10 and CIFAR100 – the same number of queries DFME and DFMS-HL use when reporting their classification results. Additionally, the first paragraph of section 4.2, which we expand in part 1 of our response to reviewer aGyH, compares the query counts and computation requirements of the different data-free model extraction methods evaluated. This details the theoretical improvement to query efficiency our optimization brings when compared with DFME.
> > > > >
> > > > > Furthermore, we would like to note that the classification results reported in Table 1 are the converged (saturated) accuracies of the different models (supplementary Figure 1 graph) – suggesting that if these methods were given an infinite number of queries, they would still not be able to fully bridge the 4% gap between the student and target model accuracy. We contribute the gap between accuracies of a data-free method with unlimited queries and the target model, to effectiveness of the method in exploring the input space. However, we agree that substantially improving the query efficiency of data-free model stealing methods is one of the primary open questions in this area.
> > > > >
> > > > > Taking values from the data plotted in the supplemental Figure 1 graph, we provide the table below to more clearly show the query efficiency of our method. The table notes the number of queries (x1 million) required to reach different accuracies on CIFAR-10 under the soft-label setting:
> > > > > | Accuracy | 75% | 80% | 85% |
> > > > > |---|---|---|---|
> > > > > | DFME | 4.89 | 7.02 | 11.56 |
> > > > > | DFMS-SL | 5.89 | 8.21 | 10.98 |
> > > > > | Dual Students | 3.57 | 5.54 | 6.43 |

---

### Official Review · Reviewer_vQa1 · 2022-10-27

**Confidence:** 4
**Correctness:** 3
**Technical Novelty And Significance:** 3
**Empirical Novelty And Significance:** 3
**Recommendation:** 8

**Clarity, Quality, Novelty And Reproducibility:**

The idea is novel. The writing needs improving as many sentences are too long to read, and some grammar mistakes should be corrected as well.

**Strength And Weaknesses:**

Pros:
1. The introduced additive student model provides an alternative way to update generator which looks reasonable and convincing.
2. The derived gradients are experimentally demonstrated closer to the true gradient (computed by target model) than DFME which only uses a single student.
3. The experiments showed that DS can approximate the target model more accurately than the baselines across various tasks.

Cons:
1. Is it possible that two student behaves like two experts in MoE. Let’s say two students work in a complementary way. Will this be risky for simply using the average output during inference? Any experimental observation about this concern?
2. The authors mentioned that Hong el al. maximized the confidence of the student output. According to my experience, this insight might be helpful to learn a data distribution that is close to the real training data. Has this method been included as a baseline?
3. Following 2, I felt that data-free model stealing has some overlap with the topic of black-box inverse attack which aims to steal the training samples. The authors should discuss their relations. Particularly, according to the presented gradient error (fig 2), I suspected that the generated data by DS or DFME does NOT looks similar to the true training samples.
4. I agreed that the proposed formulation may benefit the training of generator. However, the claim of exploring diverse training data has not been well supported. Any further evidence about it?
5. It would be perfect if some theoretical analyses are provided to support the distance of Eq. 7 is minimized by DS.
6. Please check the equation number, some of which are not properly referred.

**Summary Of The Paper:**

This paper proposed a Dual Student (DS) to mimic the behavior of a black-box model without any prior about input data. The main advantage of DS over existing work lies in that it allows a naturally differentiable process for the training of generator.

**Summary Of The Review:**

This paper is overall good. The writing needs improving as many sentences are too long to read, and some grammar mistakes should be corrected as well. I expect the authors’ response to my above comments.

---

> ### Author Response · Authors · 2022-11-10
> **Collection of requested results along with explanations and notes of changes made to the paper (part 1 of 2)**
>
> Thank you for your review of our paper. We’ve collected results, fixed minor corrections, expanded the requested discussion topic, and have revised longer confusing sentences in our paper. Please let us know what additional changes we can make to strengthen our paper and if you have any other questions or concerns.
>
> 1. Our method is more akin to an ensemble technique rather than a mixture of experts because the training data is the same across students. Although the generator is optimized to generate training samples on which the students disagree, at a high level this brings the students closer together instead of pushing the students into separate domains. Expanding our Dual Student results from Table 1, below are the individual student accuracies along with moving average accuracies and the accuracies from combining outputs in an ensemble (table 1 reported results). Although there are some cases where the ensemble reduces the performance of the best performing model (see MNIST hard, FashionMNIST hard), for the most part taken an ensemble results in higher accuracies.
> |Dataset 	| Target Accuracy	| Labels 	| Student1 	| Student2 	| Student1 MA 	| Student2 MA 	| Ensemble	|
> |-----------|-----------|-----------|-----------|-----------|---------------|---------------|-----------|
> |MNIST 		| 99.66		| Soft 		| 99.32 	| 99.36		| 99.35 	 	| 99.35 		| 99.36 	|
> |MNIST 		| 99.66		| Hard 		| 99.26 	| 99.21 	| 99.22 		| 99.18 		| 99.25 	|
> |FashionMNIST 	| 93.84		| Soft 		| 91.18 	| 90.73 	| 91.16 		| 91.09 		| 91.17 	|
> |FashionMNIST 	| 93.84		| Hard 		| 80.64 	| 80.15 	| 80.44 		| 80.29 		| 80.53 	|
> |GTSRB 		| 97.21		| Soft 		| 96.37 	| 96.33 	| 96.39 		| 96.37 		| 96.40 	|
> |GTSRB 		| 97.21		| Hard 		| 93.05 	| 93.03 	| 93.11			| 93.08 		| 93.20 	|
> |SVHN 		| 96.20		| Soft 		| 95.63 	| 95.52 	| 95.64 		| 95.61 		| 95.72 	|
> |SVHN 		| 96.20		| Hard 		| 95.19 	| 95.30 	| 95.28	 		| 95.38 		| 95.43 	|
> |CIFAR10 	| 95.5		| Soft 		| 91.05 	| 91.07 	| 91.04 		| 91.06  		| 91.34 	|
> |CIFAR10 	| 95.5		| Hard 		| 77.62 	| 77.80 	| 78.01 		| 78.55  		| 78.72 	|
>
> 2. We chose not to include the Hong et al. method (MEGA) as a baseline as the paper is unpublished, does not have code available, and only provides results for MNIST and Fashion-MNIST. We included this work in our related works because it is one of the first papers attempting to extend into the hard-label domain. Anecdotally, we did try adjusting the L1 loss to emphasize higher confidence scores in the student output but were unable to resolve the training stability of alternate losses, leading to worse results overall on CIFAR10. For convenience we provide the results from MEGA alongside our results from Table 1 below. Although these results are not directly comparable due to differing target models, the large difference suggests our method vastly outperforms MEGA.
> |Dataset  | Target Accuracy  | Method  | Soft-Labels  | Hard-Labels |
> |---|---|---|---|---|
> |MNIST  | 99.12  | MEGA  | 90.93  | 89.43 |
> |MNIST  | 99.66  | DS  | 99.36  | 99.25 |
> |FashionMNIST  | 91.15  | MEGA  | 48.89  | 52.04 |
> |FashionMNIST  | 93.84  | DS  | 91.17  | 80.53 |
>
> 	The attack success rates are not directly comparable as the attack parameters are not specified in the MEGA paper, however our results using an epsilon less than half the normal epsilon used (3/255 compared with standard 8/255) still beat their reported results.
>
> 3. The images generated during training in data-free model stealing do not resemble true samples, as is consistent with "Figure 11 synthetic" in the DFMS-HL paper and and "Figure 7" in the DFME paper. Although not explicitly noted, the example x shown in Figure 1 in our work is an example image generated during training. We have updated Figure 1 in our supplemental material with additional examples generated from the generator towards the end of training. (see part 2 of response)

---

> ### Author Response · Authors · 2022-11-10
> **Collection of requested results along with explanations and notes of changes made to the paper (part 2 of 2)**
>
> 3. (continued from part 1) Black-box inversion attack has quite a bit of overlap with regular model stealing attacks. In the former, the objective is to steal the training data with white-box or black-box access to the target model, while in the latter the objective is to steal the target model functionality. However, the objective of inversion attacks is much narrower than model stealing attacks. The samples from the training dataset can be viewed as a single specific end-point, whereas model functionality has a much broader space of "correct" solutions. For example, retraining a model on the same training data will result in similar performance with different weights and would be a "correct" solution for model stealing. As far as we can tell, the area of model inversion attacks has a reliance on real-world data and has yet to enter the data-free domain. When entering the black-box data-free domain, estimating an accurate target model gradient becomes very important for stealing the target model and for extracting training samples. Tangentially related is work done by L Engstrom et al. in “Adversarial Robustness as a Prior for Learned Representations”, which indirectly suggests inversion attacks become much more difficult in the data-free domain where only the target model is provided. Even in a white-box setting, standard models require special robustness training in order to be able to extract reasonable image reconstructions. This work, as well as work in adversarial attacks as a whole, suggests that there is significant distortion when passing an image through a model which makes it difficult to reconstruct the training samples in the data-free setting. Though data-free model stealing can get around this problem by generating samples beneficial to training but lacking in semantic meaning, it is not obvious how data-free model stealing approaches can be altered or extended to become inversion attacks.
>
>  	A reason the DFMS generated samples don't look real is that it is apparently not necessary to emulate the domain of real-world images for extracting the functionality of models. Similar to how adversarial examples expose vulnerabilities which cause classification models to have major instability in real-world image classification, these vulnerabilities perhaps make data-free model stealing easier. However, more work extending data-free model stealing to steal robust models would be needed to provide additional insight.
>
> 4. We use class distribution as our empirical metric for the diversity of training images. Because the target model is trained on a balanced dataset, we expect the images generated by the generator to also have a balanced class distribution. Figure 3 in the paper and Figure 2 in the supplementary compare the classes generated using Dual Students versus DFME, and show that using Dual Students results in more balanced class diversity. These figures indirectly show our generator improves sample diversity in the data-free domain. We’ve updated Figure 2 in the supplementary material with class distribution results for DFME, and it is clear that DS generates a more diverse set of classes than DFME.
>
> 5. Equation 7 is a measurement for empirical evidence that our proposed Equation 2 results in student gradients more similar to the target model than using the forward differences gradient estimation in DFME. Equations 1 through 6 provide an outline for how our proposed optimization should be more direct in bringing students closer to the target model, however the experimental evidence using the Equation 7 measurement was needed to show the benefit of our more direct optimization. In other words, the original methods like DFME may be optimizing the same objective as DS, however, Equation 7 shows that our approach is more effective. Though it would be great to show how Equation 7 is minimized using DS, it wouldn’t necessarily distinguish our method from existing methods, as the overall optimization objective in Equation 1 is similar.
>
> 6. We've rephrased the equation 4 reference and removed the reference for the student minimization (previously equation 5).
>
> Response to Summary. Inline with reviewer xQYc’s comment 1, we've reworked the latter part of the introduction to be more clear and have reworked a few other long sentences and areas of confusion. We hope these changes improve readability - though most of the paper was noted to be clear by reviewer yxtQ in section “Clarity” and reviewer aGyH in “Pro 1”, we'd appreciate any additional suggestions to further improve the writing.

---

> > ### Comment · Reviewer_vQa1 · 2022-11-24
> > **Thanks for the authors' repsonses.**
> >
> > My concerns have been almost addressed. I have some further questions:
> >
> > 1. Actually, Fig. 3 showed that Dual Student still exists an imbalanced issue. Have you ever considered adding a maximum entropy regularization (used in DFMS-HL) to resolve the issue?
> >
> > 2. Could you analyze why Dual Student performs better than DFME in terms of sample diversity from the perspective of their objectives?
> >
> > 3. I am wondering if it is possible that Dual Student converged to $S_1=S_2\neq T$ and $G$ then collapses?

---

> > > ### Author Response · Authors · 2022-12-06
> > > **Answers to further questions**
> > >
> > > Thank you for your response. We have provided explanations below - please let us know if you have any other concerns or further questions.
> > >
> > > 1. We have tried adding a few different class balancing terms to our training objective, including maximum entropy regularization. However, none improved classification performance on the CIFAR-10 dataset.
> > >
> > > 2. The shared framework for both Dual Students and DFME is to train students and a generator s.t. the worst-case generated sample still results in agreement between target and student. The key difference in objectives is that our method uses a more direct optimization via substitution. This substitution, incorporating a second student, results in a more accurate gradient estimation (Fig. 2) which in turn allows the generator to generate more challenging samples that are beneficial for training (Table 1). Regarding the sample diversity, we suspect the more accurate gradient estimation allows the generator to “find” the classes that are more difficult to traverse to. Figure 2 in the supplementary material shows the class distribution of generated images during training. We see that DFME finds around 6 different classes by epoch 100 but fails to find the other classes in the remaining 150 epochs, whereas Dual Students continues to find the more challenging classes as the model continues training. This result aligns with the gradient estimation of the methods (Fig. 2a): performance between DFME and Dual Students is relatively similar until around epoch 100, where the Dual Students gradient estimation becomes better than DFME. In summary, a better gradient estimate appears to result in better sample diversity (class distribution) and better overall classification performance.
> > >
> > > 3. We refer to Figure 2 to show that it is probably not the case that $S_1 = S_2 \neq T$, as the student gradients do not diverge from the target model. Modal collapse is a challenging issue, which is one reason the training iterations between generator-student and student-target are so biased towards student-target: for every 1 iteration of generator-student training, there are 5 iterations of student-target for the generated samples.

---

> > > > ### Comment · Reviewer_vQa1 · 2022-12-08
> > > > **Thanks for the reponses.**
> > > >
> > > >
> > > > It is a little strange that the maximum entropy regularization cannot work at all.
> > > >
> > > > Anyway, I think this is an interesting paper and my concerns have been adequately addressed. So I raise my score.

---

### Decision · Program_Chairs · 2023-01-20

**Decision:**

Accept: poster

**Justification For Why Not Higher Score:**

There remains a concern that the submission is lacking sufficient query efficiency analyses.

**Justification For Why Not Lower Score:**

The submission stands on its own given its merits (new, reasonable, and convincing approach to data-free model stealing).

**Metareview: Summary, Strengths And Weaknesses:**

The submission tackles the problem of data-free model stealing using an approach called Dual Students. It consists of a generator network and two student networks, in addition to the black-box target network. The generator network is tasked with producing model inputs for which the student network predictions disagree, and the student networks are tasked with matching the target network's output on the generator-produced examples. In contrast to previous work where the generator is tasked with maximizing the disagreement between a student network and the target network (thereby requiring to compute the target network's Jacobian), Dual Students is fully differentiable with respect to the generator's loss.

The proposed approach is evaluated in terms of student classification performance on MNIST, FashionMNIST, GTSRB, SVHN, and CIFAR10, and is compared against DFME and DFMS-HL. It is shown to outperform competing approaches in that setting. The submission also presents black-box transfer-based adversarial attack results, showing that data-free methods like the proposed approach are more successful than using a proxy model trained on the real dataset and also outperform a white-box FGSM attack.

Reviewers note that the submission proposes a reasonable and convincing idea (vQa1) which is novel (nQa1, xQYc, aGyH) and addresses an interesting problem (yxtQ). All reviewers also agree in one way or another that the empirical results presented look promising. Some reviewers had initial concerns regarding writing quality (vQa1, xQYc), the relationship to black-box inverse attacks (vQa1), the intuition behind proposed approach (xQYc), missing experimental details (xQYc), and missing CIFAR100 results (xQYc) which were addressed to their satisfaction by the authors.

Reviewers are split on acceptance, with Reviewers vQa1 and aGyH for it and reviewers xQYc against it. Reviewer xQYc remains concerned that the query efficiency results presented are insufficient, because they are restricted to one combination of baseline method, experiment setting, and dataset. Somewhat less importantly, they also remain concerned that the submission is missing evaluations under more attacks.

Ultimately, the majority opinion is that the submission's merits outweigh its flaws.

**Note From Pc:**

if the above contains the word "oral" or "spotlight" please see: "oral" presentation means -> notable-top-5% and "spotlight" means -> notable-top-25%. As stated in our emails, we are disassociating presentation type from AC recommendations

**Summary Of Ac-Reviewer Meeting:**

Reviewer vQa1 argued that the current results are enough to verify the paper's main claims. Reviewer xQYc remained concerned about the missing CIFAR100 results (which were later on provided by the authors) and the insufficient query efficiency experiments.